# Inadequate Energy Delivery Is Frequent among COVID-19 Patients Requiring ECMO Support and Associated with Increased ICU Mortality

**DOI:** 10.3390/nu15092098

**Published:** 2023-04-27

**Authors:** Mathias Schneeweiss-Gleixner, Bernhard Scheiner, Georg Semmler, Mathias Maleczek, Daniel Laxar, Marlene Hintersteininger, Martina Hermann, Alexander Hermann, Nina Buchtele, Eva Schaden, Thomas Staudinger, Christian Zauner

**Affiliations:** 1Department of Medicine III, Clinical Division of Gastroenterology and Hepatology, Medical University of Vienna, 1090 Vienna, Austriageorg.semmler@meduniwien.ac.at (G.S.); n11718197@students.meduniwien.ac.at (M.H.); christian.zauner@meduniwien.ac.at (C.Z.); 2Department of Anesthesia, Intensive Care Medicine and Pain Medicine, Medical University of Vienna, 1090 Vienna, Austria; mathias.maleczek@meduniwien.ac.at (M.M.); daniel.laxar@dhps.lbg.ac.at (D.L.); martina.hermann@meduniwien.ac.at (M.H.); eva.schaden@meduniwien.ac.at (E.S.); 3Ludwig Boltzmann Institute for Digital Health and Patient Safety, Medical University of Vienna, 1090 Vienna, Austria; 4Intensive Care Unit 13i2, Department of Medicine I, Medical University of Vienna, 1090 Vienna, Austria; alexander.hermann@meduniwien.ac.at (A.H.); nina.buchtele@meduniwien.ac.at (N.B.); thomas.staudinger@meduniwien.ac.at (T.S.)

**Keywords:** critical care, medical nutrition therapy, COVID-19, ECMO

## Abstract

Background: Patients receiving extracorporeal membrane oxygenation (ECMO) support are at high risk for malnutrition. There are currently no general nutrition guidelines for coronavirus disease 2019 (COVID-19) patients during ECMO therapy. Methods: We conducted a retrospective analysis of COVID-19 patients requiring venovenous ECMO support at a large tertiary hospital center. Nutrition goals were calculated using 25 kcal/kg body weight (BW)/day. Associations between nutrition support and outcome were evaluated using Kaplan–Meier and multivariable Cox regression analyses. Results: Overall, 102 patients accounted for a total of 2344 nutrition support days during ECMO therapy. On 40.6% of these days, nutrition goals were met. Undernutrition was found in 40.8%. Mean daily calorie delivery was 73.7% of calculated requirements, mean daily protein delivery was 0.7 g/kg BW/d. Mean energy intake of ≥70% of calculated targets was associated with significantly lower ICU mortality independently of age, disease severity at ECMO start and body mass index (adjusted hazard ratio: 0.372, *p* = 0.007). Conclusions: Patients with a mean energy delivery of ≥70% of calculated targets during ECMO therapy had a better ICU survival compared to patients with unmet energy goals. These results indicate that adequate nutritional support needs to be a major priority in the treatment of COVID-19 patients requiring ECMO support.

## 1. Introduction

Malnutrition is a common feature in coronavirus disease 2019 (COVID-19) patients requiring admission to the intensive care unit (ICU) [1,2,3]. A high prevalence of gastrointestinal intolerance and a hypermetabolic state with higher energy targets have been hypothesized to contribute to malnutrition in this patient population [4,5,6,7,8,9,10]. Recent data show that adequate energy delivery was associated with decreased ICU mortality in critically ill COVID-19 patients without extracorporeal membrane oxygenation (ECMO) support, highlighting the importance of sufficient nutrition support [11].

Since the outbreak of the COVID-19 pandemic, the use of venovenous (vv) ECMO has become an established rescue therapy for patients with severe respiratory failure and its use has increased significantly [12,13]. Despite tremendous advances in the management of patients receiving ECMO support, these patients still represent some of the most severely ill, requiring highly specialized intensive care treatment [14,15]. They are more likely to have a prolonged stay in the ICU [14,15]. It is assumed that the acute illness combined with the ECMO therapy itself causes significant inflammation, leading to increased protein catabolism. Whether a loss of vital nutrients occurs within the ECMO circuit itself is subject of debate [16,17]. The situation is aggravated by the fact that indirect calorimetry (the gold standard for the determination of energy expenditure) cannot be used in ECMO patients to guide energy delivery [18,19]. Consequently, critically ill patients receiving ECMO support are at high risk for the development of malnutrition, which is associated with increased morbidity and mortality [19,20,21,22,23,24,25,26].

Despite the constantly increasing number of ECMO patients, there are only scarce data concerning optimal nutritional support in this specific patient population. Nutrition deficits are common in ECMO patients [18,20,21,27,28,29]. In a cohort of non-COVID-19 ECMO patients, insufficient energy and protein supply occurred in 28.3% of observed nutrition support days during extracorporeal therapy [21]. Unfortunately, there are currently no specific nutrition guidelines concerning the optimal dose of nutrition for patients requiring ECMO therapy. Most recommendations for the nutritional management of both COVID-19 and ECMO patients refer to existing guidelines and suggest commencing EN within 48 h after ICU admission at low doses and advancing to the energy target over the following 3 days [30,31]. Specifically, ECMO support itself does not constitute a barrier for medical nutrition therapy, and enteral nutrition (EN) is generally feasible, well tolerated and safe [22,24,25,28,32,33,34]. Parenteral nutrition (PN) should only be used in patients with contraindications for EN or if sufficient EN is not tolerated [30,31].

The primary aim of the study was the description of the nutrition support practices in a large single tertiary center cohort of COVID-19 patients during ECMO support and the evaluation of the feasibility, safety, tolerance and adequacy of medical nutrition therapy for these patients according to the current “ESPEN guidelines on clinical nutrition in the intensive care unit” [30]. Finally, we wanted to investigate a possible association between nutritional adequacy during ECMO therapy and ICU mortality.

## 2. Materials and Methods

We conducted a retrospective observational study of COVID-19 patients receiving ECMO support at a large tertiary center in Vienna (Medical University of Vienna). During the first three waves of the pandemic, a total of six ICUs (medical and surgical) provided ECMO support solely for COVID-19 patients. Adult patients requiring vvECMO support due to COVID-19-associated acute respiratory distress syndrome (ARDS) between March 2020 and May 2021 were included in this study. To evaluate the nutrition support practices during ECMO therapy, the observation period was set from the first day of ECMO support until the day before ECMO decannulation or death. This study was approved by the local ethics committee of the Medical University of Vienna (ethic vote number: 2024/2020) and performed in accordance with the latest version of the Declaration of Helsinki.

### 2.1. Nutrition-Related Data

All patients in the study received nutrition support according to the ICU’s standard operating procedures, aiming to start nutrition support via nasogastric tube as soon as possible after admission to the ICU and favoring EN over PN whenever possible. Nutrition goals were set by the treating physicians, estimating daily energy and protein requirements using simple weight-based equations [30].

Daily nutrition-related-data for each patient were collected during ECMO, starting from day 1 after cannulation until the day before end of ECMO support or death. Data on nutrition intake (including energy and protein from EN, PN, intravenous glucose and propofol) and insulin requirements were assessed during 24 h periods. Adequacy of energy and protein delivery was defined according to current ESPEN guidelines [30]. Adequate delivery of nutrition was defined as 70–100% of the calculated target of energy for that day, with underfeeding defined as <70% and overfeeding >100%. Energy targets were calculated using 25 kcal/kg actual body weight (BW)/day for each day during ECMO therapy. For obese patients (BMI ≥ 30 kg/m^2^), adjusted BWs according to ESPEN guidelines were used [30]. Protein delivery is indicated as g/kg actual BW/d for every patient.

The routes of feeding (i.e., EN, PN, EN + PN, gastric tube, duodenal tube and jejunal tube) were evaluated for each day. As soon as the patient received any number of calories through a particular route of feeding during the defined 24 h period, this route was counted as valid for this day. In addition, we analyzed how many calories and proteins were supplied by each route. The frequency of prokinetic therapy (i.e., erythromycin, metoclopramide, naloxegol and methylnaltrexone) as well as metabolic (hypertriglyceridemia = triglycerides ≥ 350 mg/dL; hyperglycemia = glucose ≥ 200 mg/dL) and gastrointestinal complications (gastric residual volume (GRV) of >500 mL/24 h) was assessed for each day on ECMO and nutrition support.

### 2.2. Data Collection

Data were extracted from the patient data management system (IntelliSpace Critical Care and Anesthesia, Philips, Amsterdam, The Netherlands), which is used in all ICUs at the Medical University of Vienna, enabling a digital and prospective documentation of crucial patient characteristics including gender, age, height, weight, BMI, vital signs, comorbidities/underlying disease and laboratory tests. It also provides complete information about fluid balances, therapies and medication, nutrition and extracorporeal life support including ECMO, renal replacement therapy and mechanical ventilation.

The severity of critical illness and the extent of organ dysfunction were calculated within the first 24 h upon ICU admission and directly prior to ECMO therapy using the simplified acute physiology score (SAPS II) and sequential organ failure assessment (SOFA) score. Furthermore, we collected data on ICU length of stay (LOS) as well as ICU mortality as outcome parameters.

### 2.3. Statistical Analysis

Quantitative parameters are presented as median with interquartile range (IQR) for non-normally distributed data or as mean ± standard deviation for normally distributed data, qualitative parameters are presented as absolute numbers with relative frequencies (%). The Kolmogorov–Smirnov test was performed to test for normal distribution of quantitative variables. The primary outcome was adequacy of energy (specified as % of calculated requirements) and protein supply (specified as g/kg BW/d) during ECMO support as defined by ESPEN guidelines [30]. Predefined secondary outcome parameters were ICU mortality and gastrointestinal/metabolic complications.

In order to analyze energy and protein delivery during ECMO support, different calculations were performed:-Overall daily energy and protein delivery = mean ± standard deviation of calorie delivery (% of requirements according to ESPEN guidelines) and protein delivery (g/kg BW/d) of all patients during the entire observation period. These calculations were used to provide an overview of the nutrition support of our patient population and to make comparisons with already published data more feasible.-Time course of daily mean energy and protein delivery = mean ± standard deviation of calorie and protein delivery for each day during ECMO therapy. These data were used to analyze changes in nutrition supply over time.-In order to evaluate the effects of medical nutrition therapy on ICU survival, we divided our patient population into two groups according to the adequacy of energy supply (mean calorie delivery ≥70% and <70% of calculated requirements over the course of ECMO therapy). For this reason, calorie and protein delivery for each patient over the course of ECMO support was calculated as = mean ± standard deviation of calorie and protein delivery of each patient over the time course of ECMO support. The time period between the start of ECMO and its end, or death, was used for the calculation of mean energy and protein delivery for each patient.

In order to evaluate changes in nutrition support over the course of ECMO therapy, we calculated energy and protein delivery during the first 3 days, the first week (day 1–7) and the second week (day 8–14) of ECMO therapy, respectively.

In the case of normal distribution, metric variables were compared between the groups using *t*-test. To identify differences in baseline characteristics Pearson’s chi-square or Fisher’s exact tests were used for comparison of categorical variables, as appropriate. The Mann–Whitney U or Wilcoxon signed-rank tests were used for non-parametric variables, as appropriate.

The probability of ICU survival was calculated by the product limit method of Kaplan and Meier. Differences among various subgroups of patients concerning ICU survival were determined by log rank test. We considered *p*-values < 0.05 statistically significant.

Univariable and multivariable Cox regression analysis was used to evaluate parameters associated with ICU survival. In order to adjust for severity of the underlying disease, SAPS II was used. The overall predictive ability of investigated models was assessed by Harrel’s C index and compared among each other by using the likelihood ratio test. The relationship between energy/protein delivery and ICU survival was graphically assessed using restricted cubic spline analysis.

All statistical analyses were performed using IBM SPSS Statistics 27 (IBM, New York, NY, USA), R 4.2.2 (R Core Team, R Foundation for Statistical Computing, Vienna, Austria) and GraphPad Prism 8 (GraphPad Software, San Diego, CA, USA).

## 3. Results

We included 102 patients (73 men; 71.6%) who received vvECMO support for COVID-19-associated ARDS at the Medical University of Vienna between March 2020 and May 2021. The baseline characteristics are depicted in Table 1. The frequency of comorbidities can be found in Appendix A. At ICU admission, the median age of the study population was 57 years (50–62 years) with a BMI of 29 kg/m^2^ (26–35 kg/m^2^). Almost 50% of the patient population fulfilled the criteria for obesity (BMI > 30 kg/m^2^). Median SOFA score and SAPS II upon ICU admission at the Medical University of Vienna were 8 (7–9) and 42 (37–49), respectively. At the start of ECMO therapy, the median SOFA score and SAPS II were 8 (7–9) and 40 (34–46), respectively. We report a median duration of ECMO support of 20 days (11–31 days). The median ICU LOS was 35 days (22–57 days). The ICU mortality was 41.2% (42 patients).

Differences in the baseline characteristics and comorbidities according to ICU survival are shown in Table 1 and Appendix A. ICU survivors were significantly younger and had lower SAPSII scores at admission and ECMO start compared to non-survivors. ICU survivors had a longer ICU LOS compared to non-survivors (44 vs. 27 days; *p* = 0.0023). We did not find any differences in the frequency of comorbidities between both subpopulations.

### 3.1. Data on Nutrition Support during ECMO Therapy

Daily mean calorie delivery increased within the first days of ECMO support, followed by a plateau at 70–80% of the calculated targets (Figure 1A). Daily mean protein delivery remained very low, reaching a plateau at approximately 0.7 g/kg BW/d on day 5 (Figure 1B).

Mean daily calorie delivery accounted for 73.7% (±29.1%) of calculated requirements, mean daily protein delivery was 0.7 g/kg BW/d (±0.4 g/kg BW/d; Table 2). Feeding was most frequently performed via EN (mean daily energy delivery 51.0% (±34.6%) of requirements; mean daily protein delivery 0.5 g/kg BW/d (±0.4 g/kg BW/d)) followed by PN (mean daily energy delivery 14.2% (±23.9%) of requirements; mean daily protein delivery 0.2 g/kg BW/d (±0.4 g/kg BW/d); Table 2). Propofol was also found to be a significant source of energy delivery, accounting for a mean daily calorie supply of 8.6% (±7.7) of calculated requirements. As seen in Appendix A, a small but constant increase in energy and protein delivery during the first two weeks could be detected.

The 102 patients accounted for a total of 2344 nutrition support days. The distribution of energy and protein delivery during ECMO therapy is depicted in Table 3 and Figure 2. Energy goals were achieved on 952 (40.6%) of these days. Energy delivery was inadequate on 1392 (59.4%) days, including undernutrition on 956 (40.8%) days and overnutrition on 436 (18.6%) days. The number of days exceeding protein delivery of ≥0.7 g/kg BW was 1187 (50.6%). Indeed, in only 114 days (4.9%) the ESPEN recommendation of 1.3 g/kg BW protein delivery was reached.

The most common route of feeding was EN, accounting for a total of 1516 (64.7%) days, followed by EN + supplemental PN (sPN) with 567 (24.1%) days and total PN (tPN) with 181 (7.7%) days (Table 3). No nutrition support at all was provided on 80 (3.4%) days. EN was mainly delivered via gastric tube (2171 days, 92.6%). A duodenal/jejunal tube was used on 151 days (6.4%).

In terms of metabolic/gastrointestinal complications associated with medical nutrition therapy, we report at least one episode of hyperglycemia on 1306 (55.7%) nutrition support days. The median insulin requirements were 22.5 I.E./d (0.0–50.0). Hypertriglyceridemia was present on 801 (34.2%) nutrition support days (Table 3). The mean GRV in 24 h was 128 mL (±232 mL), a GRV ≥500 was documented on 196 (8.4%) days. One or more prokinetic drugs were used on 1247 (53.2%) days.

### 3.2. Nutrition Support and Outcome

Differences in nutrition support practices during ECMO support according to ICU survival are depicted in Table 2 and Table 3. In general, ICU survivors received significantly more calories compared to ICU non-survivors (76.3% vs. 69.8% of calculated requirements; *p* < 0.0001) and had significantly more nutrition support days with adequate energy delivery (42.3% vs. 38.0%; *p* = 0.0394). ICU non-survivors showed a higher proportion of nutrition support days on which underfeeding occurred (46.8% vs. 37.2%; *p* < 0.0001). In contrast, overnutrition was more common in ICU survivors. ICU non-survivors showed a significantly higher contribution of PN to total energy and protein delivery compared to ICU survivors (Table 2). Consistent with these findings, tPN or EN + sPN was administered on a higher number of days in non-survivors. EN alone was used less often (503 days, 53.7%) in ICU non-survivors compared to ICU survivors (1013 days, 71.9%; *p* < 0.0001). We did not find any differences in energy and protein supply over the course of the first two weeks of ECMO support (Appendix A). There was a statistically significant difference concerning the mean GRV between ICU non-survivors and survivors (146 vs. 116 mL; *p* = 0.0079). In addition, ICU non-survivors exhibited more days with GRV ≥ 500 mL per day.

In order to evaluate the effects of medical nutrition therapy on ICU survival, we divided our patient population into two groups according to the adequacy of energy supply (mean calorie delivery ≥70% and <70% of calculated requirements over the course of ECMO therapy). Differences in the basic characteristics and nutrition support practices between these groups are depicted in Appendix A. Patients with a mean calorie delivery ≥70% of the calculated requirements received more calories and protein. They had a significantly longer ECMO runtime (26 vs. 14 days; *p* < 0.0001) and a longer ICU LOS (51 vs. 27 days; *p* < 0.0001). As assessed by Kaplan–Meier survival estimation and the log rank test, a mean calorie delivery of ≥70% of calculated targets was associated with significantly lower ICU mortality (Figure 3A).

We evaluated parameters associated with ICU mortality by performing uni- and multivariable Cox regression analyses. As shown in Table 4, a mean energy intake of ≥70% of calculated targets not only decreased the risk for ICU mortality in univariable analysis (HR: 0.395 (CI 0.197–0.794), *p* = 0.009), it also proved to be independently associated with lower ICU mortality in a multivariable Cox regression analysis (aHR: 0.372 (CI 0.182–0.760), *p* = 0.007) after adjusting for age (aHR: 1.094 (CI 1.031–1.162), *p* = 0.003), disease severity (SAPS II) at ECMO start (aHR: 1.009 (CI 0.968–1.051), *p* = 0.677) and BMI (aHR: 1.071 (CI 1.019–1.125), *p* = 0.007, Harrel’s C of the overall model: 0.739 (standard error = 0.044)). Importantly, adding information on mean energy intake of ≥70% of calculated targets to the model improved the overall predictive ability (likelihood ratio test vs. model without mean energy intake: *p* = 0.005).

Mean protein delivery ≥0.7 g/kg BW/d was also found to be associated with improved ICU survival but did not reach statistical significance after adjusting for age, disease severity (SAPS II) at ECMO start and BMI using multivariable Cox regression analysis (Appendix A).

Finally, using multivariable restricted cubic spline analysis, adjusting for age, BMI and SAPSII at ECMO start, we found that the risk for ICU mortality decreased almost linearly with an increase in mean caloric and protein delivery (Appendix A).

## 4. Discussion

This study presents data from 102 patients who required ECMO therapy for COVID-19 ARDS, showing that undernutrition is common and associated with higher ICU mortality.

### 4.1. Undernutrition Is Frequent among ECMO Patients

ICU patients are considered a high-risk population for undernutrition, making medical nutrition therapy an important component of intensive care treatment [35,36]. ECMO patients are among the sickest in the ICU, exhibiting an increased metabolism and high protein consumption associated with long ICU stays and extended rehabilitation [12,14,15]. However, due to the limited data concerning nutrition support during ECMO therapy, there are currently no specific nutrition guidelines for these patients.

In our study, inadequate energy supply (under + overnutrition) occurred in 59.4% of the observed nutrition support days. The average daily energy delivery was 73.67% of the calculated requirements, indicating a high proportion of unmet energy goals. Our results are in contrast to currently available data on nutritional support in vvECMO patients [21,37]. MacGowan et al. reported a median energy delivery of 89.8% of calculated targets during their observation period, corresponding to an overall adequate energy delivery [21]. However, undernutrition still occurred on nearly one third of nutrition support days [21]. Likewise, another recent study by Hardy et al. investigating COVID-19 patients with and without ECMO showed similar rates of adequate nutrition delivery in the subgroup of ECMO patients [37]. The differences of these two studies from our results might be related to the higher disease severity in our patient population as indicated by a higher SOFA score. It is known that a higher severity of critical illness contributes to inadequate nutrition supply due to a higher frequency of (metabolic and gastrointestinal) complications and necessary interventions/procedures [38,39,40,41,42]. Another contributing factor may be an association with COVID-19 disease. The strong inflammation during COVID-19 infection was thought to cause a hypermetabolic state with higher energy targets [7,8,9,10]. However, recent studies using indirect calorimetry suggest that COVID-19 patients have similar energy expenditures compared to non-COVID-19 patients [9]. Therefore, the metabolic changes and consequently the specific nutritional needs during COVID-19 remain to be elucidated.

### 4.2. Protein Intake

Current ESPEN guidelines recommend a protein intake of 1.3 g/kg BW/day for critically ill patients [30]. We report a mean daily protein delivery of 0.7 g/kg BW. Guideline-conform protein intake was only achieved on 4.9% of the observed nutrition support days, which constitutes a much lower rate as compared to the aforementioned studies [21,37]. Both studies calculated the protein requirements using a minimum of 1.2 g/kg BW/day. Nevertheless, the average protein delivered was 84.7% [21] and 94.4% [37] of calculated requirements.

This large gap may in part be explained by an inadequate nutrition supply. We find crucial differences in daily practice with regard to the applied nutrition support between our data and the previous published studies [21,37]. In both studies, medical nutrition therapy was guided and monitored by dietitians [21,37]. In the case of unmet protein targets, protein supplements were used to reach adequate protein supply [21,37]. At our institution (Medical University of Vienna), we do not have routine dietetic assistance in the ICU to assess the adequacy of energy and protein delivery. In addition, we administer EN formulas with a standard protein content (3.8–4.1 g/100 mL). The PN formulas used exhibit higher concentrations of protein but are only used when EN is contraindicated or insufficient. Therefore, the low protein delivery in our patient population may be caused by the relatively low protein content of our EN formulas [43,44,45].

### 4.3. Association with Mortality

There are conflicting data concerning the association between ICU mortality and the adequacy of nutrition supply for ECMO patients. Only one study by MacGowan et al. analyzed the outcome and nutrition therapy in vvECMO patients [21]. The authors show that better energy and protein delivery was not associated with ICU mortality. These results are in contrast to our findings. In our study, undernutrition was associated with an increased risk for ICU mortality after adjusting for age, disease severity at ECMO start and BMI, highlighting a crucial role of nutrition in ECMO patients. However, definite causality needs to be elucidated in further, preferably interventional, trials. Based on our data, we cannot exclude that malnutrition at least partly reflects a higher degree of critical illness, which is associated with a higher frequency of metabolic and gastrointestinal complications [38,39,40,41,42]. Poor gastrointestinal tolerance to enteral feeding due to ischemia, high-dose sedation, neuromuscular blockade, opioid use and prone positioning translates into increased volumes of GRV and remains the main reason for the discontinuation of EN [19,27,29]. The ESPEN guidelines suggest a GRV of > 500 mL in 6 h in order to delay enteral feeding. We report a mean GRV of only 128 mL in 24 h, still leaving a majority of patients with undernutrition, indicating that EN was probably stopped or reduced inadequately. The vicious circle continues when adequate nutrition supply can only be achieved with PN, which is associated with higher infection rates [46,47]. Interestingly, in our study cohort, ICU non-survivors had a higher contribution of PN to nutrition supply. One explanation could be that ICU survivors exhibited more days in a stable condition, making EN more feasible. This hypothesis is supported by a significantly longer ICU LOS and ECMO runtime in ICU survivors. In addition, as indicated by the smaller mean GRV and fewer days with GRV ≥500mL, ICU survivors had a better gastrointestinal tolerance compared to ICU non-survivors.

However, it remains unclear which strategy should be favored to reduce GRV and provide adequate enteral nutrition. In our study, more than 80% of patients received prokinetic drugs during the course of ECMO therapy. Prokinetic drugs were often used in a prophylactic manner and not only in cases of high GRV or obstipation. Therefore, it cannot be assumed that prokinetic drug use is correlated with gut dysfunction. Overall, the placement of duodenal or jejunal tubes was rare. This might be explained by the associated risk for iatrogenic bleeding complications in anticoagulated patients during ECMO. However, there are currently no data analyzing the usage of postpyloric feeding tubes in ECMO patients.

### 4.4. Limitations

Due to the retrospective character of this study, only limited data on reasons for inadequate nutrition supply and feeding interruptions were available, which needs to be considered a major limitation. This likewise includes missing data, especially regarding gastrointestinal and metabolic complications.

We only collected and analyzed data concerning medical nutrition therapy and ECMO during the ICU stay at our institution. Many patients were referred from other hospitals specifically for ECMO therapy. In this respect, some patients had been in the ICU for a longer time period before our study’s observation period, with prior administration of supportive therapy including nutrition supply. These factors have not been taken into account for our analysis. However, we regard the commencement of ECMO therapy as a crucial point during treatment in the ICU. In this context, we considered it reasonable to exclude nutrition data prior to ECMO start in order to analyze the effects of ECMO support on medical nutrition therapy.

Our data show that a recommended protein delivery of 1.3 g/kg BW/d was almost never achieved at our institution. These results weaken the generalization of our data and limit the comparison to other studies with higher protein delivery.

Additionally, energy requirements were calculated by simple weight-based equations and the adequacy of energy delivery was defined according to ESPEN guidelines [30], as indirect calorimetry cannot be used during ECMO support. Therefore, it is possible that energy targets have been over- or underestimated in our study. Although there are two proposed protocols for measuring energy expenditure during ECMO support, these protocols are still experimental and have not been evaluated for routine clinical use [48,49].

The fact that ICU non-survivors had a significantly shorter ICU LOS might introduce a bias into the analysis of energy adequacy and mortality, as they might have had a lower chance of receiving adequate nutrition than survivors. However, since we did not find any significant differences between ICU survivors and non-survivors in nutrition support during the first 14 days of ECMO support (see Appendix A), this bias may only apply to later phases of ECMO support. These data also indicate that ICU non-survivors and survivors were fed with the same intention irrespective of disease severity, at least at the beginning of ECMO therapy. However, we cannot exclude the fact that medical nutrition therapy was stopped intentionally in patients with very poor prognosis. In order to minimize this bias, we therefore calculated the mean calorie delivery of each individual patient over the time course of ECMO support. By doing so, the influence of low/no medical nutrition therapy on the average energy and protein supply at the end of life is kept as low as possible.

## 5. Conclusions

In this study of 102 COVID-19 patients requiring ECMO support, we found that inadequate energy and protein delivery is common during extracorporeal therapy. Undernutrition, defined by <70% of total energy requirement, was independently associated with increased ICU mortality. Because of the retrospective study design, we cannot demonstrate causality, but our data suggest that adequate medical nutrition therapy may be essential in the management of COVID-19 patients requiring ECMO support in order to improve patient outcome. However, there are still many open questions, including optimal energy and protein requirements as well as gastrointestinal intolerance, warranting further, preferably prospective and interventional evaluations.

## Figures and Tables

**Figure 1 nutrients-15-02098-f001:**
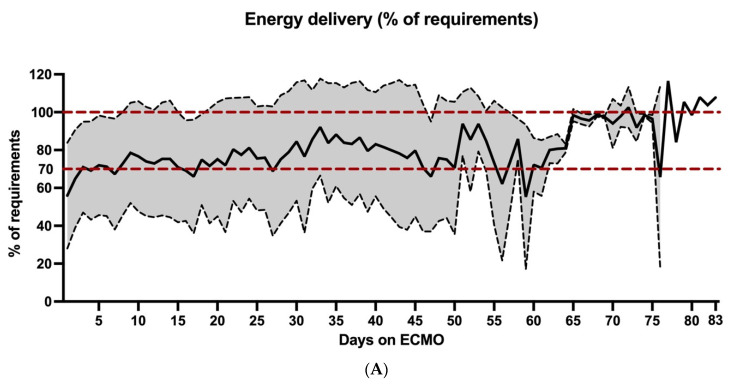
Daily mean energy and protein delivery for each day during ECMO support. Daily mean energy (% of requirements) and protein (g/kg BW/d) intake over the course of ECMO support. The black line indicates mean values, the shaded grey area the standard deviation. The horizontal red dotted lines in (**A**) illustrate adequacy of energy delivery (70–100% of requirements) according to ESPEN guidelines. The horizontal red dotted lines in (**B**) illustrate the ESPEN recommendations of 1.3 g/kg BW protein delivery per day.

**Figure 2 nutrients-15-02098-f002:**
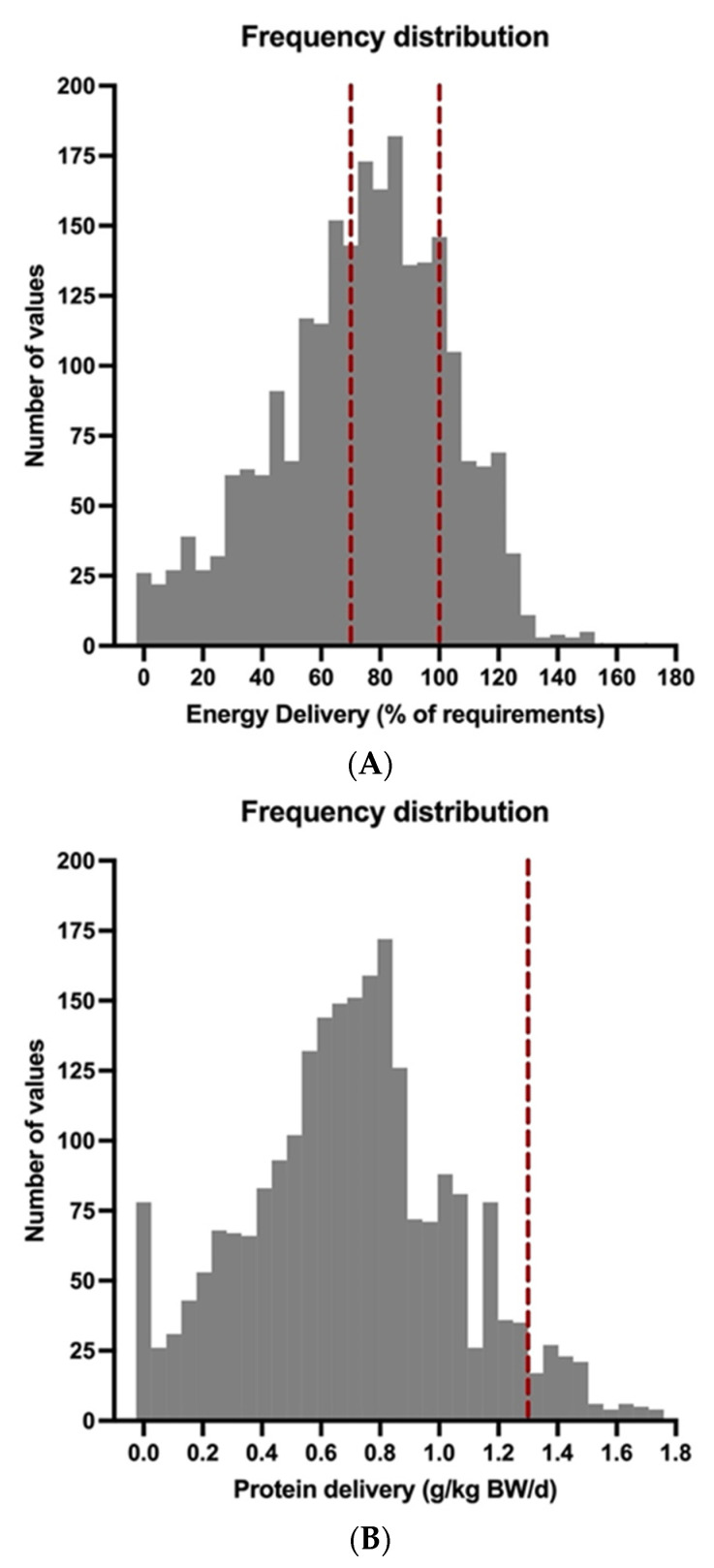
Distribution of energy and protein intake during ECMO support. Distribution of energy (**A**) and protein intake per day during ECMO support. The whole study population accounted for a total of 2344 nutrition support days. Each value in this figure represents the nutrition delivery of one nutrition support day. The vertical red dotted lines in (**A**) margin the values of adequate energy (70–100% of requirements) intake according to ESPEN guidelines. The vertical red dotted lines in (**B**) illustrate the ESPEN recommendations of 1.3 g/kg BW protein delivery per day.

**Figure 3 nutrients-15-02098-f003:**
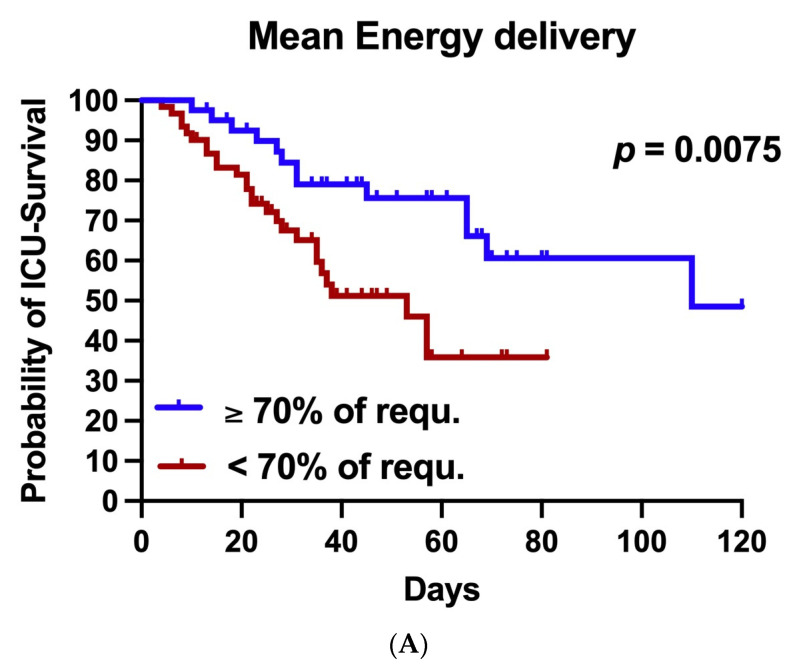
Adequacy of nutrition delivery and ICU survival in COVID-19 patients receiving ECMO therapy. The patient population was divided into two subgroups according to (**A**) mean calorie delivery ≥70% or <70% of calculated requirements or (**B**) mean protein delivery ≥0.7 or <0.7 g/kg BW/d over the course of ECMO therapy. ICU survival was calculated according to the method of Kaplan and Meier. Both mean calorie delivery ≥70% of calculated requirements (*p* = 0.0075 by log rank test) and mean protein delivery ≥0.7 g/kg BW/d (*p* = 0.0441 by log rank test) were related to an increased ICU survival in COVID-19 patients receiving ECMO support.

**Table 1 nutrients-15-02098-t001:** Basic characteristics according to ICU mortality.

Basic Characteristics	Overall	ICU Surv.	ICU Non-Surv.	*p*=
Number of Patients, No. (%)	102 (100)	60 (58.8)	42 (41.2)	
Age (years), median (IQR)	57 (50–62)	55 (48–60)	61 (57–66)	<0.0001
Male, No. (%)	73 (71.6)	40 (66.7)	33 (78.6)	0.2650
Weight (kg), median (IQR)	90 (80–100)	89 (80–100)	90 (80–105)	0.8826
BMI (kg/m^2^), median (IQR)	29 (26–35)	29 (26–34)	30 (27–36)	0.6809
SOFA at admission, median (IQR)	8 (7–9)	7 (7–8)	8 (7–9)	0.2531
SOFA at ECMO start, median (IQR)	8 (7–9)	8 (7–8)	9 (8–11)	<0.0001
SAPSII at admission, median (IQR)	42 (37–49)	40 (34–46)	45 (40–52)	0.0024
SAPSII at ECMO start, median (IQR)	40 (34–46)	39 (32–44)	43 (38–51)	0.0218
ECMO duration (days), median (IQR)	20 (11–31)	18 (12–31)	20 (9–31)	0.8776
ICU LOS (days), median (IQR)	35 (22–57)	44 (26–64)	27 (15–40)	0.0023

Abbreviations: Surv., survivors; Non-Surv., non-survivors; No., number; %, percent; IQR, interquartile range; kg, kilogram; m^2^, square meter; BMI, body mass index; SOFA, Sequential Organ Failure Assessment Score; SAPSII, Simplified Acute Physiology Score II; ECMO, extracorporeal membrane oxygenation; ICU, intensive care unit; LOS, length of stay.

**Table 2 nutrients-15-02098-t002:** Daily energy/protein delivery over the course of ECMO therapy according to ICU mortality.

Nutrition Data	Overall	ICU Surv.	ICU Non-Surv.	*p*=
Daily calorie del. (% of requ.) overall, mean ± Std.	73.677 (29.1)	76.3 (28.2)	69.8 (30.0)	<0.0001
Daily calorie del. (% of requ.) from EN, mean ± Std.	51.0 (34.6)	56.0 (33.7)	43.4 (34.5)	<0.0001
Daily calorie del. (% of requ.) from PN, mean ± Std.	14.2 (23.9)	11.6 (22.7)	18.0 (25.1)	<0.0001
Daily calorie del. (% of requ.) from prop., mean ± Std.	8.6 (7.7)	8.7 (7.6)	8.4 (7.9)	0.2867
Daily protein del. (g/kg BW/d) overall, mean ± Std.	0.7 (0.4)	0.7 (0.3)	0.7 (0.4)	0.1650
Daily protein del. (g/KG BW/d) from EN, mean ± Std.	0.5 (0.4)	0.5 (0.4)	0.4 (0.4)	<0.0001
Daily protein del. (g/kg BW/d) from PN, mean ± Std.	0.2 (0.4)	0.2 (0.4)	0.3 (0.4)	<0.0001

Abbreviations: ICU, intensive care unit; Surv., survivors; Non-Surv., non-survivors; del., delivery; requ., requirements; Std., standard deviation; EN, enteral nutrition; PN, parenteral nutrition; prop., propofol; g, gram; kg, kilogram; BW, actual body weight; d, day.

**Table 3 nutrients-15-02098-t003:** Nutrition support practices during ECMO therapy.

Nutrition Support Practices	Overall	ICU Surv.	ICU Non-Surv.	*p*=
Total number of potential nutrition support days	2344	1408	936	
Days with calorie del. 70–100% of requ., No. (%)	952 (40.6)	596 (42.3)	356 (38.0)	0.0394
Days with calorie del. <70%of requ., No. (%)	956 (40.8)	522 (37.1)	434 (46.4)	<0.0001
Days with calorie del. >100% of requ., No. (%)	436 (18.6)	290 (20.6)	146 (15.6)	0.0024
Days with protein del. ≥0.7 g/kg BW/d, No. (%)	1187 (50.6)	709 (50.3)	478 (51.1)	0.7360
Days with protein del. ≥1.3 g/kg BW/d, No. (%)	114 (4.9)	54 (2.3)	60 (6.4)	0.0058
Days with EN, No. (%)	1516 (64.7)	1013 (71.9)	503 (53.7)	<0.0001
Days with PN, No. (%)	181 (7.7)	79 (5.6)	102 (10.9)	<0.0001
Days with EN + PN, No. (%)	567 (24.1)	274 (19.5)	293 (31.3)	<0.0001
Days with no nutrition support, No. (%)	80 (3.4)	42 (3.0)	38 (4.1)	0.1649
Days with GRV ≥500mL, No. (%)	196 (8.4)	97 (6.9)	99 (10.6)	0.0018
Days with prokinetic therapy, No. (%)	1247 (53.2)	739 (52.5)	508 (54.3)	0.3984
Days with ≥1 episode of hyperglycemia, No. (%)	1306 (55.7)	733 (52.1)	573 (61.2)	<0.0001
Days with hypertriglyceridemia, No. (%)	801 (34.2)	466 (33.1)	335 (35.8)	0.1770

Abbreviations: ICU, intensive care unit; Surv., survivors; Non-Surv., non-survivors; del., delivery; requ., requirements; No., number; %, percent; EN, enteral nutrition; PN, parenteral nutrition; prop., propofol; g, gram; mg, milligram; kg, kilogram; BW, actual body weight; d, days.

**Table 4 nutrients-15-02098-t004:** Univariable and multivariable analyses of prognostic factors for ICU survival according to adequacy of energy delivery.

Parameter	Univariable		Multivariable	
	HR (95%CI)	*p*-Value	aHR (95%CI)	*p*=
Mean Energy del. ≥70% vs. <70% of requ.*	0.395 (0.197–0.794)	0.009	0.372 (0.182–0.760)	0.007
Age	1.068 (1.026–1.112)	0.001	1.094 (1.031–1.162)	0.003
SAPSII at ECMO Start	1.037 (1.005–1.069)	0.023	1.009 (0.968–1.051)	0.677
BMI	1.042 (0.996–1.089)	0.074	1.071 (1.019–1.125)	0.007

* Patients were divided in two groups according to the adequacy of energy supply (mean calorie delivery ≥70% and <70% of calculated requirements over the course of ECMO therapy). Harrel’s C of the overall model: 0.739 (standard error = 0.044). Abbreviations: del., delivery; requ., requirements; vs., versus; Sequential Organ Failure Assessment Score; SAPSII, Simplified Acute Physiology Score II; ECMO, extracorporeal membrane oxygenation; BMI, body mass index; HR, hazard ratio; aHR, adjusted hazard ratio; CI, confidence interval.

## Data Availability

The data that support the findings of this study are available from the corresponding authors upon reasonable request.

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
