# Peer review of "Inadequate Energy Delivery Is Frequent among COVID-19 Patients Requiring ECMO Support and Associated with Increased ICU Mortality"

_nutrients, 2023, doi:10.3390/nu15092098_

Round 1

Reviewer 1 Report

Thank you very much for allowing me to review the paper Inadequate energy delivery is frequent among COVID-19 pa-2 tients requiring ECMO support and associated with increased 3 ICU mortality.

I would like to congratulate the authors for this work as it meets all the requirements of a good research paper.

An introduction with a precise justification of the study problem leads the reader to the relevant objectives of the study.

A correct choice of material and methods in order to be able to adequately measure the stated objective.

A correct presentation of results in order to determine the achievement of the stated objective.

The right discussion with the comparison of results with previous work including the differences found with possible reasons.

The presentation of the limitations is important as it clears up the reader's doubts regarding the results.

Likewise, a concise and clear conclusion leaving the necessary avenues of research to go deeper into a clear problem such as hospital malnutrition, in this specific case, in patients subjected to ECMO therapy, and its implication in mortality, drawing attention to a little studied aspect, the inclusion of dietician-nutritionists in critical patient units to evaluate nutritional care.

I reiterate my congratulations to the authors.

Reviewer 2 Report

The study aimed to investigate the association between nutritional adequacy during ECMO therapy and ICU mortality. The results may be able to add some contributions to the guideline of clinical treatment. Here are my comments.

1. The authors are suggested to add C indexes and use it to estimate the fitness of the Cox hazard model.

2.Line 277, the authors added a study and demonstrate it was similar to the current study. Then, they stated that there are differences between the studies and gave their explanation. This might have made readers confused about the differences.

3.EN, PN, GRV and prokinetic therapy are related to the survival. Why were they not analyzed in the COX multivariated model?

4.In order to adjust for severity, the authors used SAPS as a proxy for adjusting the effects of comorbidities/underlying disease. However, SAPS score includes only three comorbidities (i.e., AIDS, metastasis cancer and blood caner), some major comorbidities such as CVD, DB which are related to mortality are excluded. Further statements are needed.

5.The depiction of p value in the footnote of Table S1 is unclear.

6.The first parameter in Table S4 is unclear.

7.Line 369, Should Table S2 be replaced with Table S1?
